# Treatment of Rare Mutations in Patients with Lung Cancer

**DOI:** 10.3390/biomedicines9050534

**Published:** 2021-05-11

**Authors:** Tarek Taha, Rasha Khoury, Ronen Brenner, Haitam Nasrallah, Irena Shofaniyeh, Samih Yousef, Abed Agbarya

**Affiliations:** 1Oncology Institute, Technion Faculty of Medicine, Rambam Health Care Campus, 8 HaAliyah HaShniyah Street, 3525408 Haifa, Israel; t_taha@rambam.health.gov.il (T.T.); h_nasrallah@rambam.health.gov.il (H.N.); 2Bnai-Zion Medical Center, Oncology Institute, Technion Faculty of Medicine, 47 Golomb Avenue, 3339419 Haifa, Israel; rasha.khoury@b-zion.org.il; 3The Edith Wolfson Medical Center, Oncology Institute, 62 Halohamim Street, 5822012 Holon, Israel; ronenbr@wmc.gov.il; 4MyBiotics Institute, 2 Yitzhak Modai Street, 7608804 Rehovot, Israel; irenashofaniyeh@gmail.com; 5Oncology Community Unit, Clalit Health Service, 15 Marj Ibn Amar Street, 1603701 Nazareth, Israel; samihys@clalit.org.il

**Keywords:** lung cancer, non-small cell lung cancer, gene alterations, mutations, targeted therapy, pharmaceutical agents, clinical trials, response rate, progression-free survival, side effects

## Abstract

Lung cancer is a worldwide prevalent malignancy. This disease has a low survival rate due to diagnosis at a late stage challenged by the involvement of metastatic sites. Non-small-cell lung cancer (NSCLC) is presented in 85% of cases. The last decade has experienced substantial advancements in scientific research, leading to a novel targeted therapeutic approach. The newly developed pharmaceutical agents are aimed towards specific mutations, detected in individual patients inflicted by lung cancer. These drugs have longer and improved response rates compared to traditional chemotherapy. Recent studies were able to identify rare mutations found in pulmonary tumors. Among the gene alterations detected were mesenchymal epithelial transition factor (MET), human epidermal growth factor 2 (HER2), B-type Raf kinase (BRAF), c-ROS proto-oncogene (ROS1), rearranged during transfection (RET) and neurotrophic tyrosine kinase (NTRK). Ongoing clinical trials are gaining insight onto possible first and second lines of medical treatment options intended to enable progression-free survival to lung cancer patients.

## 1. Introduction

Lung cancer (LC) is the most commonly diagnosed malignancy in the world, with more than 2 million new cases a year. It is the leading cause of death, responsible for approximately 19% of cancer-related deaths, more than breast, prostate and colon malignancies combined [1,2]. There are two major subtypes of LC, based on histology examination. Non-Small Cell Lung Cancer (NSCLC) is the most prevalent lung cancer subtype, accounting for 85% of cases. NSCLC is further histologically classified as non-squamous (70%) and squamous (25%). The remaining 15% are Small Cell Lung Cancer (SCLC) [3]. The majority of LC cases are diagnosed in an advanced stage, and thus have poor prognosis [4]. In recent years, a profound progression in the fields of diagnosing and treating LC has been achieved. In the field of diagnosis, the progress is attributed to our understanding of the molecular pathogenic pathways leading to this type of cancer, while in the field of treatment, the progress is related to the introduction of Immunotherapy and Biological treatments for patients with LC. While immunotherapy acts via augmenting the response of the immune system against cancer cells, the biological treatments promote their activity by blocking signal transduction pathways. The discovery of targetable genetic alterations has thoroughly shifted the treatment paradigm of metastatic lung cancer. This change has affected the means of drug administration, tailored to individual patients’ specific classification of tumor mutation, and thus improved response rates. Intravenous chemotherapeutic agents prescribed to all NSCLC patients, resulting in limited response rates (RR), have been replaced by targeted therapies. The latter are mostly orally administered, and are given to a specific group of patients, characterized by certain genetic alterations. These treatments provide profound and durable responses, expressed by longer progression-free survival (PFS) rates [5].

The paradigm shift is also true regarding the use of Immune Checkpoint Inhibitors (ICIs). While ICIs today serve as the backbone treatment of Stage IV LC with no targetable mutations, their use in the treatment plan of LC with driver mutation is controversial. Treating oncogenic-driven metastatic NSCLC with ICIs causes unacceptable toxicities before and after administering a Tyrosine Kinase Inhibitor (TKI) while yielding to only low to modest RRs in the first line setting [6].

At present, we are able to perform molecular testing on cancer cells derived from the tumor tissue itself or from circulating tumor DNA (ctDNA), collected from plasma. This approach has improved our ability to obtain more tumor cells for molecular profiling. The advantage of using ctDNA is the availability of the sample. Obtaining tumor tissue cells is done by drawing blood, which is less invasive than other procedures, such as bronchoscopy and lung puncture. Another advantage of ctDNA analysis is that it represents the entire tumor genome, rather than the tissue, because of the tumor heterogeneity. One crucial disadvantage of using the ctDNA is the reduced threshold sensitivity to the mutations and the clinical meaning of the very low level of mutations. Advanced molecular testing with Hybrid Capture Next Generation Sequencing (NGS), enables us to discover a wide range of common and less common mutations. The most common targetable driver alterations in LC are mutations in Epidermal Growth Factor Receptor (EGFR) 10–20% [7], and in Anaplastic Lymphoma Kinase (ALK) 3–7% [8]. Other less common mutations include: Mesenchymal Epithelial Transition factor (MET), Human Epidermal growth factor 2 (HER2) and B-type Raf Kinase (BRAF) genes, and rearrangements in c-ROS proto-oncogene (ROS1), rearranged during transfection (RET) and neurotrophic tyrosine kinase (NTRK) genes (Figure 1). Today, it is recommended to perform comprehensive molecular profiling during the first stage of diagnosis, as a part of disease staging, together with other molecular testing for programmed death-ligand 1 (PD-L1), and Micro-Satellite Instability (MSI). Additionally, it is also recommended to repeat molecular profiling after disease progression on first-line treatment with TKIs, in patients with EGFR and ALK mutations. In this article, we will discuss and review the milestone trials of drug development for the rarer gene alterations and their current standard of care.

## 2. Rare LC Mutations

### 2.1. ROS1 Gene

#### 2.1.1. Description

ROS1 is a tyrosine-kinase receptor not normally expressed in the lung, and its role in humans is undefined [9]. Rearrangements in ROS1 gene happen in 1–2% of NSCLC, usually in females, younger patients, light smokers or non-smokers [10,11]. ROS1 gene mutations cause the dysregulation and constitutive activation of growth and survival pathways such as mitogen-activated protein kinase (MAPK), PI3K/AKT/mTOR pathway, and STAT3 [12,13]. Approximately 36% of treatment-naïve ROS1 LC patients are diagnosed with brain metastases [14].

#### 2.1.2. Detection Methods

ROS1 gene rearrangements can be detected by fluorescence in situ hybridization (FISH), immunohistochemistry (IHC), polymerase chain reaction (PCR) and NGS.

#### 2.1.3. Pharmaceutical Agents

Crizotinib, a tyrosine kinase inhibitor (TKI) was developed and used for ALK and MET alterations. It received approval by the FDA for treating ROS1-rearranged NSCLC patients, after Phase I study PROFILE 1001 showed a 72% overall response rate (ORR), 90% disease control rate (DCR), and 19.2 months of median PFS (mPFS) [15]. Even though, phase II studies showed more modest mPFS (10–15 months) results, Crizotinib was the standard of care for many years [16,17]. The December 2020 National Comprehensive Cancer Network (NCCN) guidelines recommend treating ROS1 rearranged NSCLC with Crizotinib, Ceritinib and Entrectinib as first-line therapies (Table 1).

Ceritinib, another potent ALK inhibitor, showed profound ORR (62%), DCR (81%) and mPFS (19.3 months) in a Korean phase II study. However, these results were demonstrated only in Crizotinib-naïve patients, thereby approving Ceritinib only as a first-line therapy [18].

Entrectinib, a TKI that selectively targets ROS1/ALK/NTRK, was shown to be more potent than Crizotinib in preclinical studies [19]. An analysis of several early studies showed substantial ORR (41%) and mPFS of 19 months. Entrectinib crosses the blood brain barrier (BBB) and has significant activity in the central nervous system (CNS) metastasis. Due to the fact that CNS is a common first site of progression in ROS1 NSCLC patients on Crizotinib, Entrectinib serves as an attractive option for second-line treatment [14,20]. One study showed that Entrectinib has 55% Intracranial ORR and more than 12 months duration of response. Patients without CNS metastasis had mPFS of 26 months [21].

Additional agents, although not yet recommended by NCCN, were investigated. Lorlatinib was studied on 69 ROS1 rearranged LC patients, showing ORR of 62% and mPFS of 21 months in Crizotinib-naïve patients. Additional advantages of Lorlatinib are its good intracranial activity and efficacy in pretreated patients. Lorlatinib had 64% intracranial ORR in Crizotinib-naïve and 50% in Crizotinib-treated patients. Since Crizotinib-refractory patients have limited treatment options, Lorlatinib potentially serves as an important next-line targeted agent [22]. Crizotinib has the lowest effect on intracranial disease due to its limited penetrance to the BBB, while Ceritinib, Entrectinib, and, as mentioned above, Lorlatinib, have good BBB penetrance and demonstrable intracranial efficacy.

No disease-limiting toxicities (DLTs) were reported for Crizotinib. However, the majority of treated patients suffered mild visual impairments.

All TKIs used to treat NSCLC ROS1 mutations cause gastrointestinal (GI) side effects. The highest incidence of symptoms was reported as a consequence of Ceritinib therapy (40%), while Lorlatinib and Entrectinib caused less frequent GI symptoms [23]. A specific adverse effect, occurring in 40% of Lorlatinib-treated patients, is peripheral neuropathy (Figure 2). NCCN guidelines recommend Lorlatinib and Entrectinib as second-line therapies (Table 1).

#### 2.1.4. Role of Immunotherapy

Very little is known about the effect of ICIs on ROS1-mutated NSCLC. Most of it is derived from the IMMUNOTARGET registry, which was created to retrospectively examine the real-world data of ICIs effect on targetable NSCLC tumors [24]. The use of monotherapy ICI resulted in 17% RR, but the cohort included only seven patients. Due to this limitation, no mPFS results could be retrieved. Twelve ROS1-mutated patients were treated by ICI and enrolled in a Korean study. They demonstrated 25% RR [25]. Some case reports indicated a certain efficacy of ICIs as advanced lines [26]. Due to the lack of reliable data and many effective targeted therapy options, we suggest treating ROS1-mutated NSCLC with targeted therapies, even in the advanced lines.

#### 2.1.5. Mechanisms of Resistance

The main known mechanisms of drug resistance are associated with resistance to Crizotinib as a first-line therapy. The resistance may occur due to mutations in the ROS1 tyrosine kinase domain or due to the activation of factors in the signaling cascade [27]. The most common mutations are G2032R, D2033N, S1986Y/F, gatekeeper L2026M, and L1951R. As mentioned above, Lorlatinib has good activity in Crizotinib-refractory patients. Other agents known to have activity, although not used due to limited data or toxicities, are Cabozantinib and Repotrectinib [9].

### 2.2. RET Gene

#### 2.2.1. Description

RET is a growth factor receptor of the glial cell line-derived neurotrophic factor receptor [28]. RET rearrangements/fusions incidence in NSCLC is 1–2%, resulting in independent homodimerization of the receptor and to constitutive kinase activity. These alterations differ from RET mutations, which are more common in Medullary thyroid cancer [29]. Alterations in RET usually occur in young females, light smokers and non-smokers [11,30].

#### 2.2.2. Detection Methods

RET mutations are detected by FISH or NGS, but can also be detected by IHC, Reverse transcription PCR (RT-PCR) and circulating tumor-DNA (ctDNA) [31].

#### 2.2.3. Pharmaceutical Agents

Primarily, RET-altered NSCLC were treated by drugs targeting multiple alterations (Cabozantinib, Lenvatinib, RXDX-105, etc.) [32,33,34], resulting in a modest responses and poor tolerability due to off-targeted activity. As a consequence, two drugs were specifically designed to target RET: BLU-667 (Pralsetinib), LOXO-292 (Selpercatinib). Studies were conducted using these two drugs, which were highly selective RET TKIs and have CNS activity [35,36].

Pralsetinib was studied in phase I/II study, ARROW, conducted on 114 patients. ORR was 57% in 87 patients previously treated with platinum-based chemotherapy, while 70% ORR was reported in 27 treatment-naïve patients. Pralsetinib can cause grade 3–4 neutropenia and hypertension. Of note, in the same trial, 7% of participants discontinued the treatment due to adverse events.

Selpercatinib was tested on 144 patients, demonstrating ORR of 64% in 105 previously treated patients and 85% in 39 treatment-naïve patients. Eleven patients with CNS metastasis had remarkable intracranial response—91% [37]. Selpercatinib may cause grade 3–4 transaminasitis and hypertension. Only 1.7% of patients discontinued treatment with this drug due to adverse events.

The NCCN guidelines recommend Selpercatinib, Pralsetinib, Cabozantinib and Vandetanib as 1st and advanced treatment lines in RET-mutated NSCLC (Table 1).

#### 2.2.4. Role of Immunotherapy

According to a retrospective study which included 29 RET-altered NSCLC patients, ICIs have some activity on RET-rearranged NSCLC as an advanced treatment line. Of the 29 patients, 16 were treated with ICI and had shorter time-to-treatment discontinuation period compared to the 13 patients who were treated with Non-ICI treatment [38]. In the IMMUNOTARGET registry, one of the 16 patients with RET-rearranged NSCLC responded to ICI (16% RR with 2.1 months of mPFS). Other registry (GFPC 01-2018) showed 38% RR for ICI based on 9 RET-altered patients [39]. The majority of RET-altered patients in the registries mentioned above had surprisingly low PD-L1 expression rates, which may negatively influence the results. We suggest treating RET-altered NSCLC with targeted therapies and chemotherapy, especially taking into consideration the data showing the good activity of Pemetrexed-based chemotherapy in RET-altered NSCLC [40].

#### 2.2.5. Mechanisms of Resistance

RET resistance could occur due to gatekeeper mutations or changes in the RAS/MAPK signaling pathway, such as mutations in NRAS, KRAS or MET amplification, EGFR overexpression and AXL overexpression [41]. There are at least fourteen known resistance mutations, resulting in different resistance profiles to multiple TKIs, with Nintedanib being the less affected TKI [42]. Lately, G810 solvent front mutation was described as a recurrent mechanism of resistance to selective RET inhibition with Selpercatinib [43,44].

### 2.3. BRAF Gene

#### 2.3.1. Description

BRAF mutations promote cell proliferation and survival by phosphorylating and activating the MAPK/extracellular signal-regulated kinase (ERK) downstream signal pathways. BRAF Mutations are found in 1–5% of NSCLC [45,46], equally divided between V600 (Valine replaced by other amino acid at position 600) and non-V600 [47], with V600E (Glutamic acid replacing Valine) being the most common V600 mutation, which occurs at the level of T1799 transversion in exon 15. Non-V600E BRAF mutations may be either activating (i.e., G469A/V, K601E, L597R) or inactivating (i.e., D594G, G466V) [48]. The BRAF V600E mutation results in constitutive activation of the MAPK/ERK pathway, causing a 10-fold increase in BRAF activity [49]. According to previous evaluations, BRAF mutations in NSCLC were believed to be more prevalent in female patients. However, other evaluations did not statistically prove this observation [50]. BRAF mutations’ link to age and smoking status is also invariable [51,52].

#### 2.3.2. Detection Methods

BRAF mutations can be detected by IHC, RT-PCR, Sanger sequencing and NGS.

#### 2.3.3. Pharmaceutical Agents

Targeting BRAF-mutations gained impressive successes in other types of cancer, hereby shedding light on BRAF-mutated NSCLC. Unlike other targetable mutations, BRAF-mutated malignancies-focused therapy consists of addressing both the BRAF and the downstream MEK protein together. Former studies showed that targeting BRAF alone causes the development of upstream and downstream bypass pathways, such as RAS, MEK and ERK pathways and these, in turn, lead to resistance and decreased response to treatment [53,54].

A combination of Dabrafenib and Trametinib was studied in a Phase II study, recruiting 57 previously treated patients. The outcomes were 63% ORR and showed manageable safety profile [55]. Similar study was conducted on 36 treatment-naïve patients. The combination of Dabrafenib and Trametinib demonstrated 64% ORR and mPFS of 10.9 months [56] on treatment-naïve patients. This therapy combination reported side effects were pyrexia (in more than 30% of cases) and, less frequently, nausea, vomiting, diarrhea and asthenia.

The NCCN guidelines recommend a combination of Dabrafenib and Trametinib as first and subsequent lines of therapy for BRAF mutation (Table 1). 

#### 2.3.4. Role of Immunotherapy

The most valuable data we have, although retrospective and not necessarily positive, about the effect of ICI in oncogenic-driver-mutated NSCLC is in BRAF-mutated NSCLC. In addition to the IMMUNOTARGET and GFPC 01-2018 registries, retrospective studies from Israel, Italy and the United States evaluated the activity of ICIs on BRAF-mutated NSCLC tumors and reported RRs that range between 10% and 30% with 3 months of mPFS [57,58,59]. Results were better in smoking patients and in non-V600E mutated tumors. Overall, ICIs were shown to be a reasonable tool in the armamentarium of treating BRAF-mutated NSCLC. Due to the limited options of targeted drugs for BRAF-mutated NSCLC, we suggest using ICIs as a second-line treatment after progression on Anti-BRAF and Anti-MEK drugs.

#### 2.3.5. Mechanisms of Resistance

Anti-BRAF drugs revolutionized the treatment paradigm in BRAF-mutated tumors, especially in Melanoma. Almost 50% of metastatic melanoma patients have BRAF-mutant tumors; treating these patients with Anti-BRAF drugs produces magnificent RRs of 50–60% but its Achilles heel is the short duration of treatment before resistance and progression. Several resistance mechanisms are known—expression of CRAF kinases, elevated expression of COT, BRAF V600E amplification, NRAS upregulation, aberrant splicing of BRAF, PTEN loss, persistent activation of insulin-like growth factor 1 receptor, platelet-derived growth factor receptor and EGFR [48]. Some of these resistance mechanisms can be avoided by combining anti-BRAF drugs with anti-MEK drugs. Although second-generation anti-BRAF drugs (such as PLX8394) showed some activity in resistant tumors [48,60], at present, there are no reliable second-generation drugs.

### 2.4. NTRK Gene

#### 2.4.1. Description

NTRK are tyrosine kinase receptors, which play a physiologic role in the development of the nervous system [61]. The NTRK family genes (NTRK1, NTRK2, NTRK3) comprise of three transmembrane proteins (TRKA, TRKB, TRKC).

NTRK gene fusions in NSCLC were first identified in a study generating NGS tests on lung tumor samples [62]. Further studies revealed that NTRK fusions occur generally in fewer than 1% of NSCLC cases [63,64]. In contrast with other targetable driver mutations, NTRK-rearranged lung cancers are not limited to specific subgroups and are not associated with specific clinical characteristics. NTRK-rearrangements are described in the literature across different histology types, in all sexes, ages and smoking [65].

#### 2.4.2. Detection Methods

NTRK fusions can be detected by IHC, FISH, RT-PCR and NGS.

#### 2.4.3. Pharmaceutical Agents

Early phase clinical trials tested several TKIs as potential treatments for NTRK rearrangements. The most promising TKIs were shown to be Larotrectinib and Entrectinib.

Larotrectinib is a selective TRKA/B/C inhibitor, primarily tested in a Phase I study which enrolled 55 patients with different NTRK-rearranged tumor types, including four patients with NSCLC. The study reported 75% ORR and 80% DCR, granting Larotrectinib the FDA and European Medicine Agency (EMA) approval. Updated results from the expanded patient cohort of the same study (153 patients) confirmed the preliminary results demonstrating 79% ORR, and mPFS of 28 months [66].

Entrectinib inhibits TRKA/B/C and has inhibitory activity in ALK and ROS1 mutations. It was studied in several phase I/II trials including 74 NTRK-rearranged patients (13 NSCLC patients) which reported in an integrated analysis promising results of 63.5% ORR and 11.2 mPFS [67].

Both TKIs, Larotrectinib and Entrectinib, were shown to have potential CNS activity [68].

The most frequent reported side effects related to Larotrectinib were fatigue, elevated transaminases, constipation, dizziness, nausea and vomiting, diarrhea and pyrexia. Entrectinib most frequently causes dysgeusia, dizziness, constipation, diarrhea and fatigue. In 35% of cases, it caused grade 3–4 adverse events, with weight gain and neutropenia being the most common.

The NCCN guidelines recommend Larotrectinib and Entrectinib as first and subsequent lines of therapy in NTRK-fused NSCLC tumors (Table 1).

#### 2.4.4. Role of Immunotherapy

The IMMUNOTARGET and GFPC 01-2018 registries did not enroll any patients with NTRK alterations. Some data suggest that NTRK-fused tumors are associated with relatively high PD-L1 expression levels and a high Tumor mutational burden, suggesting that these tumors may have a good response to immunotherapy [63]. On the other hand, NTRK-fused NSCLC tumors are known to co-exist with STK11 mutations, which seem to be associated with a poor response to ICIs in KRAS mutant disease [69]. To the best of our knowledge, at present, there are no reliable data proving the efficacy or any results of ICI treatment in NTRK-fused NSCLC tumors.

#### 2.4.5. Mechanisms of Resistance

There are several known primary and acquired resistance mechanisms to TKIs targeting NTRK-altered tumors. According to one study, Ponatinib and Nintedanib could overcome a novel NTRK resistance mutation (G667C). Larotrectinib, Entrectinib, Ponatinib and Nintedanib failed to show activity against the G595R resistance mutation [70].

### 2.5. MET Gene

#### 2.5.1. Description

MET is a hepatocyte growth factor receptor with tyrosine kinase activity [71]. Alterations in the MET signaling pathway have been widely described in carcinogenic processes in several solid tumor types, including NSCLC [72]. While previously treated NSCLC patients mainly harbor MET amplification, approximately 2–4% of NSCLC treatment-naïve patients harbor a skipping mutation in exon 14 of MET [73]. Unlike the other rare mutations described above, which are mainly present in nonsquamous histology, MET mutations are harbored in 2% of squamous cell carcinoma histology [74,75]. Differing from other mutations, the incidence of MET mutations is higher in elderly, non-smoking patients [76].

#### 2.5.2. Detection Methods

MET mutations can be evaluated by differential MET exon expression, quantitative RT-PCR, RNA sequencing and NGS.

#### 2.5.3. Pharmaceutical Agents

Preliminary studies have reported that MET-mutated NSCLC may respond to treatment with MET inhibitors. A multicenter retrospective analysis demonstrated improvement in overall survival when treating MET-mutated NSCLC with MET inhibitors [77].

At present, the NCCN recommend treating MET-mutated NSCLC with Crizotinib or Capmatinib (Table 1).

Profile 1001 study recruited 18 MET-mutated NSCLC patients and treated them with Crizotinib. The oral multi-TKI protocol treatment resulted in 44% ORR. A total of 33% of the 13 patients who had amplified c-MET, responded to treatment, most of them from the high-level amplification group [78,79].

Capmatinib, a highly selective c-MET inhibitor, was tested in phase I clinical trials. The outcomes obtained ORRs between 18% and 63% in c-MET positive, cMET IHC +3 and cMET overexpressed NSCLC [80]. A phase II study GEOMETRY mono-I tested Capmatinib in different patient populations. The ORR reached 40.6% and mPFS was 5.4 months in pretreated NSCLC patients; however, in treatment-naïve patients, ORR levels were 67.9%, resulting in mPFS of 9.7 months. Capmatinib was also shown to have more than 50% Intracranial ORR (7 out of 13 patients with brain metastasis) [81]. The most common adverse events of Capmatinib are peripheral edema, nausea, vomiting and increased creatinine level. In the trial mentioned previously, 67% of patients experienced grade 3–4 toxicity.

#### 2.5.4. Role of Immunotherapy

The IMMUNOTARGET registry enrolled 36 patients with MET exon 14 skipping mutation and demonstrated 16% RR with 3.4 mPFS. A series including 24 patients who were treated with ICI as first or advanced lines showed similar results, with 17% RR and 1.9 months of mPFS [82]. The GFPC 01-2018 registry demonstrated better results, with 36% RR and 4.9 months of mPFS to 30 MET-mutated patients that were enrolled. A German study that enrolled patients with several oncogenic driver mutations concluded that patients with MET exon 14 skipping mutation responded well to therapy, with three out of eight patients partially responding and one patient with stable disease. One of the patients who partially responded had a long PFS of 115.2 weeks [83]. Another study described six MET-mutated patients with long and durable responses to ICI (18–49 months) [84]. Overall, MET exon 14 skipping mutations seem to respond well to ICI, and ICI should be a reasonable treatment strategy. These results need to be confirmed in prospective clinical trials.

#### 2.5.5. Mechanisms of Resistance

Resistance could be acquired due to on-target or off-target mechanisms. On-target acquired mechanisms could be single or polyclonal and include mutations in codons H1094, G1163, L1195, D1228, Y1230 or high levels of amplification of the MET exon 14-mutant allele. Off-target mechanisms could occur due to amplifications in KRAS, MDM2, EGFR, CDK4, HER2, HER3, BRAF, PIK3CA, loss of PTEN and mutations in KRAS [85,86,87]. Two TKIs Merestinib and Glasetinib were recently shown to have activity after acquiring resistance to Crizotinib [85].

### 2.6. HER2 Gene

#### 2.6.1. Description

Alterations in HER2 genes have been discovered in the carcinogenesis pathway of several cancers. Whereas overexpression is the most known alteration for its implication on breast and gastric cancers, affecting treatment and prognosis, mutations are more clinically relevant in the NSCLC carcinogenesis. HER2 overexpression and gene amplification are seen in NSCLC, but seem to have no clinical implication, besides indicating poorer prognostic outcomes [88,89]. HER2 mutations are present in 1–4% of NSCLC cases, mostly in never-smoking females [90,91].

#### 2.6.2. Detection Methods

To detect HER2 overexpression or amplification, IHC and FISH are usually used, since these molecular essays are cheap and easy to perform, yet practical and precise. To detect HER2 exon 20 insert mutation, NGS is usually used.

#### 2.6.3. Pharmaceutical Agents

Several studies evaluated the effect of TKIs and anti-HER2 antibodies to address HER2 alterations, without providing significant clinical outcomes. Single agents Afatinib, Dacomitinib, Lapatinib and Neratinib provided modest ORRs 0–15% in different studies involving HER2-mutated NSCLC patients [92,93,94]. Ado-Trastuzumab emtansine (T-DM1), an antibody–drug conjugate, provided ORR of 44% with 5 months mPFS in a basket trial [95]. Unfortunately, these results were not consistent in other trials studying T-DM1, and thus T-DM1 did not enter into practice. The activity of Pyrotinib, a pan-HER TKI, was addressed in a multi-center phase II trial including 60 patients, demonstrating promising ORR of 30% and 6.9 months of mPFS [96]. Recently, data from phase II ZENITH20 trial, which addresses Poziotinib in HER2-mutated NSCLC patients, were published, showing mPFS of 5.5 months and 27.8% ORR [97]. As of writing this paper (March 2021), treatment for HER2-mutated NSCLC is without established standard of care (Table 1).

#### 2.6.4. Role of Immunotherapy

Twenty-nine patients with HER2-mutant lung tumors were enrolled in the IMMUNOTARGET registry. Treatment with ICI resulted in poor RR of 7.4% with 2.5 months of PFS. The GFPC 01-2018 registry included 23 HER2-mutated NSCLC patients who were treated by ICI and demonstrated 27.3% RR (six patients), with 2.2 months of PFS. A retrospective study from Memorial Sloan Kettering Cancer Center included 26 patients and showed 12% RR [98].

#### 2.6.5. Mechanisms of Resistance

Our literature review did not reveal any reliable data regarding resistance mechanisms.

## 3. Discussion and Future Directions

NSCLC mutations targeted therapy is the new generation of pharmaceutical agents. Each drug has a certain effectiveness, expressed by response rate range, with some side effects. Ongoing trials evaluating new inhibitors for the rare gene alterations in NSCLC, discussed above, are currently being conducted (Table 2). Efforts are continuing to seek possible therapy options to offer progression-free survival for LC patients. In addition, attempts to treat other targetable alterations are being investigated. For example, mutations in Kirsten rat sarcoma viral oncogene homolog (KRAS), detected in 25–32% of NSCLC patients, comprise the most known oncogene driver mutation in NSCLC [99,100]. The vast majority of the KRAS mutations occur at codons 12 and 13. KRAS G12C mutation is the most common KRAS mutation (39%), found in 13% of overall NSCLC cases [101]. KRAS-mutant NSCLC patients are usually former or current male smokers, unlike most NSCLC driver mutations [102]. Sotorasib, an irreversible KRAS G12C inhibitor, was evaluated in the phase I/II CodeBreak 100 trial. The study enrolled 126 patients and showed an ORR of 37% and mPFS of 6.8 months. In light of these results, the FDA has recently granted Sotorasib, a priority review to treat *KRAS* G12C–mutated NSCLC patients [103]. Many other oncogenic driver mutations are being studied. FGFR3, NRG1, STK11, NF-1, TP53 are some of them. Unfortunately, we could not refer to them, as they are beyond the scope of this review.

## Figures and Tables

**Figure 1 biomedicines-09-00534-f001:**
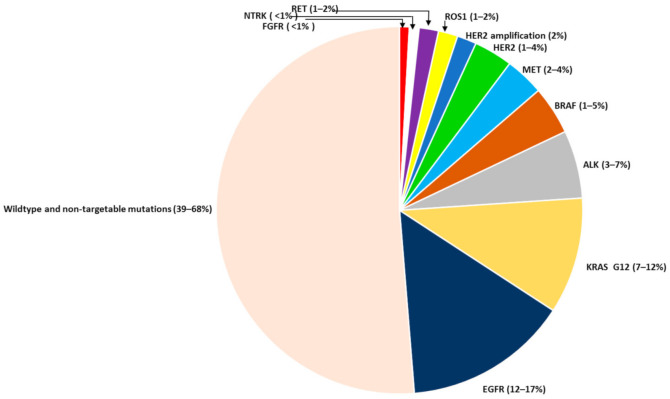
Incidence of NSCLC gene alterations.

**Figure 2 biomedicines-09-00534-f002:**
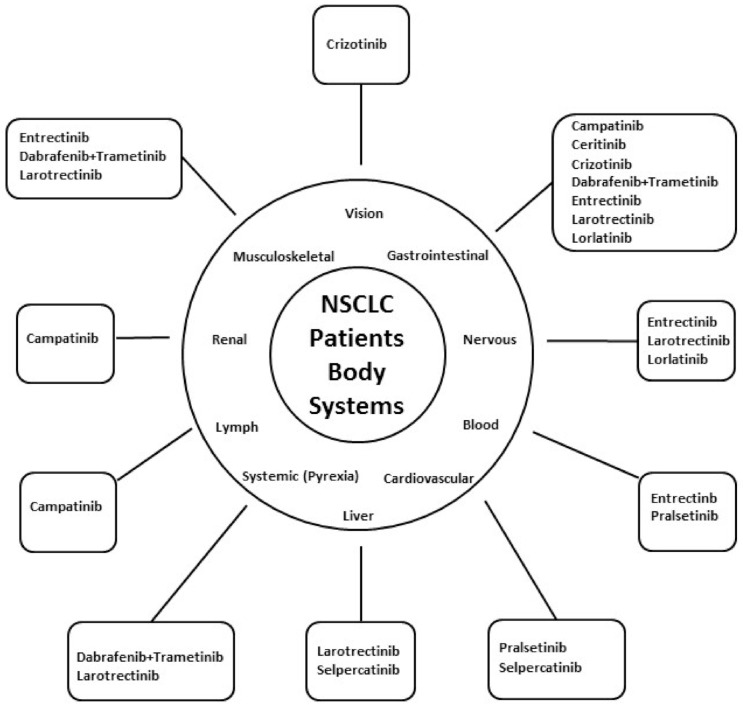
Common targeted therapies’ adverse events, categorized by systems. The figure shows the main targeted therapies causing adverse events in each of the main body systems. Clockwise: The main targeted therapy causing visual impairment is Crizotinib. Capmatinib, Ceritinib, Crizotinib, Dabrafenib + Trametinib, Entrectinib, Larotrectinib and Lorlatinib—cause Gastrointestinal adverse events. Entrectinib, Larotrectinib and Lorlatinib may cause neurological adverse events. Entrectinib and Pralsetinib are the most common causes of cytopenia. Pralsetinib and Selpercatinib could cause cardiovascular adverse events. The main causes of liver toxicity are Larotrectinib and Selpercatinib. Larotrectinib and the combination of Dabrafenib and Trametinib can cause Pyrexia. Capmatinib is the most common cause of lymphadenopathy. Capmatinib is the most common targeted therapy to cause renal toxicity. Entrectinib, Dabrafenib and trametinib and Larotrectinib may cause musculoskeletal adverse events.

**Table 1 biomedicines-09-00534-t001:** FDAapproved pharmaceutical agents to treat NSCLC gene alterations.

Agent	Target	Approved for	Date
Ceritinib	ALK	ROS1 rearranged	2014
Crizotinib	ALK, MET	ROS1 rearranged, MET mutated	2016
Entrectinib	TRK, ROS, ALK	ROS1 rearranged, NTRK fused	2019
Dabrafenib	BRAF	BRAF mutated (In combination with trametinib)	2015
Trametinib	MEK1/2	BRAF mutated (In combination with dabrafenib)	2015
Larotrectinib	NTRK	NTRK fused	2018
Capmatinib	MET	MET mutated	2020
Selpercatinib	RET	RET mutated	2020
Pralsetinib	RET	RET mutated	2020
Cabozantinib	VEGFR2, PDGFR, KIT	RET mutated	2020
Lorlatinib	ALK, ROS1	ROS1 rearranged	2018

FDA: Food and Drug Administration.

**Table 2 biomedicines-09-00534-t002:** Ongoing clinical trials to treat NSCLC mutations [103].

Gene Alterations	Pharmaceutical Agent	Clinical Trial No.
ROS1	Lorlatinib	NCT01970865
	Entrectinib	NCT02568267
	Cabozantinib	NCT01639508
	DS-6051b	NCT02279433
	TPX-0005	NCT03093116
RET	Cabozantinib	NCT01639508, NCT04131543
	Alectinib	NCT03445000, NCT03178552, NCT02183883
	TPX-0046	NCT04161391
	BOS172738	NCT03780517
BRAF	Dabrafenib + Trametinib Vemurafenib Encorafenib + Binimetinib	NCT03543306, NCT01336634 NCT04302025 NCT04526782
NTRK	Cabozantinib	NCT01639508
	LOXO-195	NCT03215511
	Repotrectinib	NCT03093116
	DS-6051b	NCT02675491
	PLX7486	NCT01804530
	Merestinib	NCT02920996
	VMD-928	NCT03556228
	MGCD516	NCT02219711
	ONO-7579	NCT03182257
MET	Crizotinib	NCT00585195, NCT02465060, NCT02499614, NCT02664935, NCT01121575, NCT00965731
	Cabozantinib	NCT00596648, NCT03911193, NCT01639508, NCT02132598, NCT03468985
	Merestinib	NCT02920996
	Glesatinib	NCT02954991, NCT02544633
	Foretinib	NCT02034097
	Capmatinib	NCT03693339, NCT02750215, NCT02468661, NCT03647488, NCT02414139, NCT01911507, NCT02323126, NCT02335944, NCT02276027
	Tepotinib	NCT01982955. NCT02864992, NCT03940703
	Savolitinib	NCT03944772, NCT03778229, NCT02117167, NCT02897479, NCT02143466, NCT02374645
	Tivantinib	NCT01251796, NCT01069757
	SAR125844	NCT02435121
	Onartuzumab	NCT02031744, NCT01519804, NCT01496742, NCT01887886
	Telisotuzumab	NCT03574753
	JNJ-61186372	NCT02609776
	Ficlatuzumab	NCT01039948
HER2	Trastuzumab-deruxtecan	NCT04644237

## Data Availability

Not applicable.

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
