# Peer review of "Treatment of Rare Mutations in Patients with Lung Cancer"

_biomedicines, 2021, doi:10.3390/biomedicines9050534_

Round 1

Reviewer 1 Report

The authors described the treatment of rare mutations in patients with lung cancer thoroughly. This review article seems to be informative and well summarized.

One thing I regret is  that this sort of reviews normally deals with many  topics, so the content seems to be narrating and sort of boring.

The addition of some figures about the characteristics of specific genes or clinical manifestations of a certain type of lung cancer would make the article more interesting. 

When it comes to Figure 2, the detail should be included in figure or legends.  

I recommend the addition of NTRK gene related lung cancer's clinical characteristics. 

Author Response

Reviewer 1:

"The authors described the treatment of rare mutations in patients with lung cancer thoroughly. This review article seems to be informative and well summarized."

Thank you for your positive feedback.

"One thing I regret is that this sort of reviews normally deals with many topics, so the content seems to be narrating and sort of boring. The addition of some figures about the characteristics of specific genes or clinical manifestations of a certain type of lung cancer could make the article more interesting."

Thank you for your feedback. Due to the nature of this review, and taking into account its' restrictions (many topics, words count, etc.), we thought that adding two figures and two tables would be sufficient.

"When it comes to Figure 2, the detail should be included in figure or legends."

Corrected. Descriptive details were added under the Figure 2.

"I recommend the addition of NTRK gene related lung cancer's clinical characteristics"

Corrected. This information was added in 2.4.1 section

Reviewer 2 Report

Comprehensive and well-organized review of a relevant topic, with helpful and concise figures/tables. I would suggest adding the citation that one of the authors themselves left in a comment. 

Author Response

Reviewer 2:

"Comprehensive and well-organized review of a relevant topic, with helpful and concise figures/tables."

Thank you for your positive feedback.

"I would suggest adding the citation that one of the authors themselves left in a comment"

Corrected. The citation was added to the text.

  • All revisions were highlighted in yellow and tracked.